# Visual Impairment Screening Assessment (VISA) tool: pilot validation

Fiona J Rowe, Lauren R Hepworth, Kerry L Hanna, Claire Howard

## ABSTRACT

**Objective** To report and evaluate a new Vision Impairment Screening Assessment (VISA) tool intended for use by the stroke team to improve identification of visual impairment in stroke survivors.

**Design** Prospective case cohort comparative study.

**Setting** Stroke units at two secondary care hospitals and one tertiary centre.

**Participants** 116 stroke survivors were screened, 62 by naïve and 54 by non-naïve screeners.

**Main outcome measures** Both the VISA screening tool and the comprehensive specialist vision assessment measured case history, visual acuity, eye alignment, eye movements, visual field and visual inattention.

**Results** Full completion of VISA tool and specialist vision assessment was achieved for 89 stroke survivors. Missing data for one or more sections typically related to patient's inability to complete the assessment. Sensitivity and specificity of the VISA screening tool were 90.24% and 85.29%, respectively; the positive and negative predictive values were 93.67% and 78.36%, respectively. Overall agreement was significant; k=0.736. Lowest agreement was found for screening of eye movement and visual inattention deficits.

**Conclusions** This early validation of the VISA screening tool shows promise in improving detection accuracy for clinicians involved in stroke care who are not specialists in vision problems and lack formal eye training, with potential to lead to more prompt referral with fewer false positives and negatives. Pilot validation indicates acceptability of the VISA tool for screening of visual impairment in stroke survivors. Sensitivity and specificity were high indicating the potential accuracy of the VISA tool for screening purposes. Results of this study have guided the revision of the VISA screening tool ahead of full clinical validation.

## BACKGROUND

Visual impairment following stroke is common and estimated to affect two-thirds of all stroke survivors.[1] There is currently no standardised protocol for screening or referral and, as a result of poor/absent screening, a considerable proportion of patients who have visual problems go unrecognised, thus receiving no advice or management.[2] There are various visual treatment options that can have a beneficial effect on vision and to general rehabilitation.[3–5] Visual impairment can have a substantial impact on quality of life including loss of confidence,

### Strengths and limitations of this study

► Iterative development process for the screening tool.
► Prospective clinical pilot validation process.
► Comparison made between naïve and non-naïve screeners.
► Acceptability of the screening assessment to stroke survivors was not captured.
► The duration of the screening assessment was not captured.

impaired mobility, inability to judge distances and increased risk of falls.[3] There is a known link between poor vision, quality of life and depression in older persons.[4 6] For these reasons, it is important that patients with visual impairment are identified by the stroke multidisciplinary team (MDT) and appropriate referral made for specialist vision assessment. It is equally important that the effects of visual impairment on functional ability are established and information is provided regarding the use of residual vision to facilitate general rehabilitation. These issues have been recognised as research priorities in the James Lind Alliance sight loss prioritisation process, in which screening and assessment of stroke survivors for visual problems is listed as a top 10 priority for research.[7]

The overall aim of this study was to develop and evaluate a Vision Impairment Screening Assessment (VISA) tool using simple established assessments of visual function coupled with detailed instructions. Our objectives were to test the VISA screen against a reference of a specialist vision assessment to determine sensitivity, specificity, predictive values and inter-rater agreement of results between the VISA screen and specialist vision assessments. A final objective was to evaluate user views on the acceptability of use of the VISA screening tool.

## METHODS
### Development

The VISA screening tool was developed following consultation with an expert panel consisting of: stroke-specialist clinical

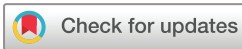

Department of Health Services Research, University of Liverpool, Liverpool, UK

**Correspondence to**
Prof. Fiona J Rowe;
rowef@liverpool.ac.uk

**Table 1** Calculations of sensitivity, specificity and predictive values

| | |
|---|---|
| *Positive* | |
| True positive, that is, visual impairment present and referred | 74 |
| False negative, that is, visual impairment present but not referred | 8 |
| *Negative* | |
| False positive, that is, visual impairment not present but referred | 5 |
| True negative, that is, visual impairment not present and not referred | 29 |
| *Output* | |
| Sensitivity (true positive/true positive+false negative) | 90.24% (95% CI 81.68% to 95.69%) |
| Specificity (true negative/false positive+true negative) | 85.29% (95% CI 68.94% to 95.05%) |
| Positive predictive value (true positive/false positive+true positive) | 93.67% (95% CI 86.78% to 97.09%) |
| Negative predictive value (true negative/false negative+true negative) | 78.38% (95% CI 64.91% to 87.66%) |

orthoptists, stroke research orthoptists, stroke survivors with visual impairment, stroke-specialist occupational therapists and neuro-ophthalmologists. The panel considered results of recent stroke/vision research studies in which multiple measures of visual function were made.[2 8] They identified the consistent vision measures across the common visual impairments occurring following stroke—those of impaired central vision, eye movement, visual field and visual inattention (the vision modality of spatial neglect).

Stroke survivors provided specific input on potential burden of these assessments to individuals, particularly when undertaken in the early acute stage post-stroke onset. Following this panel discussion, a draft screening tool was circulated along with detailed instructions compiled for each of the screening assessments, which comprised a screen of visual symptoms and observed signs, visual acuity, eye alignment and movements, visual field boundaries and visual inattention. An iterative process was followed in which the panel provided written feedback on the first and subsequent drafts of the screening tool. Feedback from both clinicians and stroke survivors guided the revision of the symptom history section to reduce the number of questions being asked and refine the question wording to remove potential ambiguity. Feedback specifically from clinicians also guided the revision of the self-guided instructions to provide more steps and detail plus to remove potential ambiguity.

The final pilot version of the VISA tool contained the same five sections as the original draft, consisting of a case history section in which visual symptoms and observed signs are documented, a visual acuity section to screen central vision at near and distance using logMAR and N-series letters, an ocular alignment and movement section to screen the presence/absence of strabismus (eye position) and eye movement problems, a visual field section to screen for peripheral field of vision by a guided confrontation method, and a visual perception section to screen for visual inattention/neglect using a triad of line bisection, cancellation task and clock drawing assessments. The VISA tool provides detailed instructions regarding correct use of the assessments required for screening. This self-directed design with the incorporation of detailed instructions as part of the tool was developed on the basis that many stroke clinicians do not have any formal eye training and may not have access to such training. Thus, the aim was to provide in-built instructions in lieu of formal training.

### Pilot validation

A prospective case cohort comparative design was used for the pilot validation clinical study. Individuals were suitable for inclusion if they were 18 years of age or older, had clinical diagnosis of stroke as defined by the WHO,[9] had the ability to agree to vision screening using verbal or non-verbal indications of agreement, did not have severe cognitive impairment preventing screening and did not decline vision screening. Our inclusion criteria were intended to be pragmatic and inclusive of as many stroke survivors as possible. The clinical study was undertaken in accordance with the Tenets of Helsinki with the National Health Service (NHS) research ethical approval.

For the purpose of this study, vision screening was undertaken with the VISA screening tool and screening was defined as the assessment of stroke survivors for the presence of reduced visual function against preset abnormality criteria. Specialist visual assessment was defined as the vision assessment undertaken by eye-trained clinicians (orthoptists and ophthalmologists) in which detection of visual impairment was coupled with formal diagnosis of the type of visual condition present.

Recruitment took place across three hospitals in which an orthoptist was a member of the core acute stroke unit MDT (as per national guidelines: Royal College of Physicians Intercollegiate Stroke Guidelines and British and Irish Orthoptic Society extended guidelines for stroke practice).[10 11]

Each stroke survivor underwent two vision assessments: the routine orthoptic specialist vision assessment (n=5 orthoptists/ophthalmologists) and the VISA screening assessment. The VISA screen was completed by medical students and orthoptists. Medical students (n=2) were chosen as screeners to represent completely naïve individuals in conducting vision screening assessments. Orthoptists (n=4) were

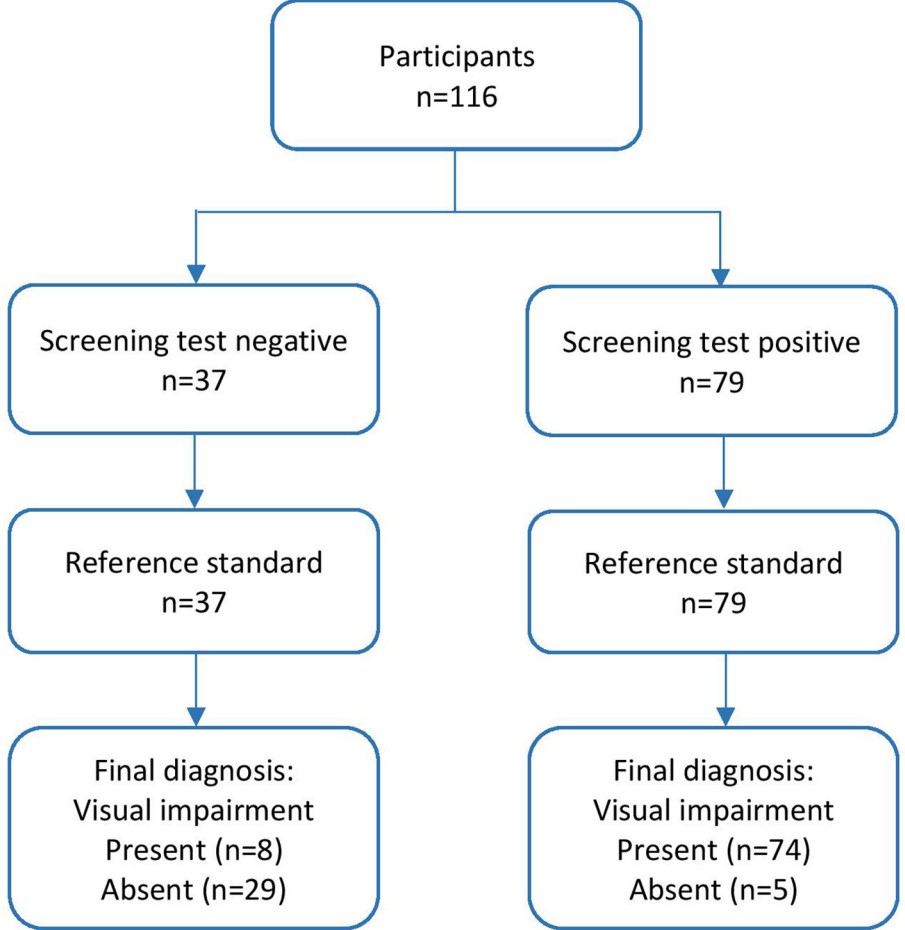

**Figure 1** Flow diagram of participant outcome for screening and full assessment.

also chosen as screeners in this pilot stage of validation to serve as a quality check of the screening tool's ability to accurately assess various aspects of visual impairment.

Routine specialist vision assessment comprised detailed diagnostic assessments of case history, visual acuity, ocular alignment and movement, visual field and visual perception. This assessment was undertaken within 24 hours (typically the same day) of the VISA screen—to minimise effect of potential recovery.

The order of the VISA screening and specialist vision assessments varied to avoid the effects of fatigue and bias towards either the screen or vision assessments. The screener and orthoptist were blinded to each other's assessments to prevent bias of assessment. The within-assessment order of testing varied for the specialist assessment. However, the order of testing within the VISA screen followed a set order of (1) case history, (2) visual acuity, (3) eye position, (4) visual field and (5) visual inattention assessments.

### Statistical methodology and sample size

Results were taken in numerical format from the referral forms completed by both the screener and the orthoptist. The specialist vision assessment was taken as the reference standard.

The primary outcome measure was presence or absence of visual impairment (defined as low vision <0.2, visual field loss, eye movement abnormality, visual perceptual abnormality) and recorded as a binary measure: yes/no for presence/absence of visual impairment. The primary outcome measure was evaluated by kappa values assessing chance-eliminated agreement between the VISA screening and specialist vision assessment results.

Secondary outcome measures were the calculation of sensitivity, specificity and predictive values. Level of sensitivity was estimated as the proportion of patients with visual impairment who are correctly identified by the screener, and the corresponding 95% CI was calculated. Level of specificity was estimated as the proportion of patients without visual impairment who are correctly identified by the screener, and the corresponding 95% CI. Further, we calculated the positive and negative predictive values for the VISA screen.

As this was a pilot validation study, we sought to include a minimum sample size of 100 subjects. This sample size is typically used for diagnostic accuracy studies, which we considered appropriate even though this was a study of screening detection rather than diagnostic accuracy.[12]

**Table 2**  Summary of agreement between the VISA screen and specialist vision assessment for referral to specialist eye services and individual components

| Element of testing | Agreement | False negative | False positive | Kappa value (95% CI) |
|---|---|---|---|---|
| Referral | 103 | 8 | 5 | 0.736 (0.602 to 0.870) |
| Near visual acuity | 93 | 10 | 7 | 0.682 (0.543 to 0.820) |
| Distance visual acuity | 94 | 8 | 3 | 0.785 (0.665 to 0.904) |
| Ocular alignment | 112 | 4 | 0 | 0.585 (0.221 to 0.949) |
| Ocular motility | 89 | 21 | 6 | 0.120 (−0.071 to 0.311) |
| Visual fields | 94 | 3 | 8 | 0.741 (0.599 to 0.884) |
| Visual inattention | 67 | 1 | 16 | 0.361 (0.144 to 0.578) |

VISA, Vision Impairment Screening Assessment.

## Process evaluation

Process evaluation for acceptability of the VISA tool during the clinical study was through a combination of feedback sheets and one-to-one interviews with screeners. Feedback sheets could be returned at any time during the study to report any issues with testing alongside obtaining clinician views based on their use of the VISA tool. Feedback sheets asked the following:

1. Are the instructions for the various tests clear?
2. Which instructions should be amended?
3. What additional instruction information/rewording do you suggest?
4. Which instructions require less information?
5. Are any tests not useful or difficult to do? (Specify.)
6. Should any other tests be added in?
7. How long does it take you to do the screen?
8. Other comments?

These questions were also asked during individual interviews. Interviews were conducted by the lead author (FJR).

Interviews and feedback sheets were transcribed verbatim and all identifying features removed. Qualitative data analysis was undertaken as an ongoing iterative process. All transcripts were systematically coded manually. A thematic approach to analysis of the qualitative data was adopted. Transcripts were coded by sentence or section and the code descriptors were derived directly from the text. A thematic approach to analysis of the qualitative data was adopted. Codes were grouped for similar content and these groups defined the key emerging themes. A modified grounded theory approach was undertaken in which themes were revised iteratively as further interviews and analysis progressed.

## RESULTS

### Completion rate

One hundred and sixteen patients with stroke received both a VISA screening assessment and a reference vision assessment over 4 months (December 2015 to March 2016). Two medical students conducted 62 of the VISA screens and 54 were screened by a team of four orthoptists. Independent specialist vision assessment was conducted by a team of four orthoptists and one ophthalmologist.

The VISA screen was fully completed by 89 patients, with the remaining 28 missing one or more elements (near vision n=4, distance vision n=6, convergence n=3, visual fields n=9, visual inattention n=28). The specialist vision assessment was fully completed by 77 patients, with the remaining 40 missing one or more elements (near vision n=3, distance vision n=9, convergence n=18, visual fields n=9, visual inattention n=23). Reasons for missing data were captured and typically related to patient's inability to complete sections of vision assessments because of impaired cognitive ability or fatigue. All patients were included even if there were missing data—missing data did not automatically result in failure for that section, thereby requiring referral. Reasons behind the failure to complete sections were always taken into consideration.

### Referral agreement

The agreement of whether to make a referral to specialist eye services based on the results of the VISA screening tool versus those from specialist vision assessment had a kappa value of 0.736 (95% CI 0.602 to 0.870).

In this pilot evaluation of the VISA screening tool, sensitivity of 90.24% and specificity of 85.29% were found. The positive and negative predictive values were 93.67% and

**Table 3**  Summary of agreement between the VISA screening tool and specialist vision assessment for referral to specialist eye services when used by a naïve versus non-naïve screener

| Screener | Agreement | False negative | False positive | Kappa value (95% CI) |
|---|---|---|---|---|
| Medical student n=62 | 51 | 7 | 4 | 0.617 (0.415 to 0.820) |
| Independent orthoptist n=54 | 52 | 1 | 1 | 0.899 (0.761 to 1.000) |

VISA, Vision Impairment Screening Assessment.

78.36%, respectively. These calculations are outlined in table 1.

Agreement was found for 103 participants (29 had no visual impairment, 74 required referral because of failed screening), outlined in figure 1. The VISA screen produced eight false negative and five false positive results. Of the false negative results, three had ocular motility problems, three had reduced distance vision, one had reduced near vision and one did not have visual fields tested during screening. For false positive results, two with visual inattention, two with visual field loss and one with both visual inattention and visual field loss were detected by screening and found not to be present by the specialist vision assessment.

### Test component agreement

The agreement for the individual components between the VISA screen and specialist vision assessments is outlined in table 2. The highest levels of agreement were produced for distance visual acuity (0.785) and visual fields (0.741). The lowest levels of agreement were produced for ocular motility (0.120) and visual inattention (0.361). Low agreement for ocular motility related to high false negatives where 21 cases (three with multiple conditions) were not detected—these comprised: nine defects of vertical movement (including four age-related restrictions, one IV[th] cranial nerve palsy and one V pattern), eight cases of nystagmus (including four end-point nystagmus), five restrictions of horizontal eye movements and four cases of reduced convergence. The low agreement with visual inattention related to false positive referrals because of failure of the patient to complete this section due to impaired cognitive ability or fatigue—rather than true presence of visual inattention.

### Naïve versus non-naïve screeners

The agreement on whether to make a referral to specialist eye services based on results of the VISA screening tool versus those from specialist vision assessment was stronger when made by a non-naïve screener (table 3). A higher rate of false positives and false negatives was found when the screener was naïve to vision testing (11 false referrals for naïve vs 2 for non-naïve screeners). The agreement on whether to make a referral to specialist eye services between the VISA screening tool and a specialist vision assessment had a kappa value of 0.736 (95% CI 0.602 to 0.870).

When used by a naïve screener the VISA screen has a sensitivity of 82.93% and specificity of 80.95%. When used by non-naïve screeners the VISA screen has a sensitivity of 97.56% and specificity of 92.31%.

### Process evaluation

Information from feedback sheets and detailed notes from interviews were compiled and grouped for type of feedback. Group themes included instruction feedback, section feedback and referral feedback.

### Instruction feedback

Screeners asked for brief instruction reminders at the top of VISA screening assessments, for example, position test chart at 3 m from the patient, cover each eye in turn, and so on. This served to act as a quick reminder for the correct procedure for that particular section of the screening tool. Clarifications were requested for the main instruction training section such that potential ambiguity was removed.

### Section feedback

In the first version, each screening section was coupled to the detailed assessment instructions. Screeners requested that all detailed instructions be merged into one training 'manual' section with the screening assessments separate. As screeners became more familiar with the tool, they used the VISA screens on their own and kept the detailed instructions elsewhere (mainly for reference) which meant there was less paperwork to be carried to the bedside assessment.

### Referral feedback

Most feedback concerned patients who were borderline on whether to refer for specialist vision assessment or not. For example, where the patient had borderline visual acuity responses—perhaps because glasses were not available—but all other visual function assessments passed the VISA screen. In other cases, the patient lacked sufficient cognitive or communication abilities rendering some VISA screens 'unsure' or incomplete. Detailed referral guidelines were compiled to guide the referral process with minimum guidance being to repeat the VISA screen 1–2 days later for borderline cases. This aimed to reduce the levels of false referrals.

### DISCUSSION

In this study, we present the VISA screening tool which encompasses screening of key visual functions affected by stroke; namely central vision, peripheral visual field, eye position/movements and visual attention, alongside ocular history. Overall, referral had sensitivity and specificity of about 90% and 80%, respectively, positive and negative predictive values of about 94% and 78%, respectively, with agreement between VISA screening and comprehensive specialist assessment of above kappa 0.7. Agreement was lowest for eye movement screening and visual inattention whereas all other individual sections showed agreement of above kappa 0.5. Low agreement in these sections related to high false positive referrals where VISA screen indicated a fail for ocular motility or visual inattention. The specialist vision assessment detected ocular motility changes which were classed as 'normal' physiological eye movement patterns such as V pattern and end-point nystagmus, and which alone would not have required referral. The detection of these physiological eye movement patterns was regarded as a positive finding within the eye movement section indicating that

the ocular motility section had proved to be sensitive to these less obvious eye movement problems. However, this section requires close monitoring in further studies to refine related training and referral guidelines. False positive referrals for visual inattention occurred where the patient failed to complete the section because of fatigue or cognitive impairment. The incomplete results were interpreted as borderline fail by screeners. Visual inattention was the last section to be completed in the VISA screen so, as a result, was likely to be most susceptible to the effects of fatigue and impaired cognition. Guidance on completing the VISA screen was therefore amended such that the more interactive components of the VISA screen (ie, visual field and visual inattention sections) were advised to be completed first in cases where cognition or fatigue could impact on screen complete; plus a repeat second screen was advised where indicated.

Process evaluation aided further refinement of the VISA screening tool and, in particular, training elements and referral guidance to add quick tips and reminders, and to remove ambiguity. Vision screening of stroke survivors by orthoptists using validated assessments has been shown to provide accurate identification of visual impairment and is easily undertaken on the stroke unit with further follow-up arranged in eye clinics as required.[13] Such orthoptic input has been reported to help prevent misdiagnosis, provide quick access to treatment of visual problems and improve response to general rehabilitation.[4 14] Orthoptists are a member of the core acute stroke MDT.[10] Despite consistent findings that inclusion of vision services within the MDT is highly beneficial, such visual assessment is not common and services are inconsistent throughout the UK. One survey showed that 45% of stroke services provided no formal vision assessment for patients with stroke.[15] A further survey of practice identified that only 7% of stroke units had a policy relating to vision assessment and management.[16] Both surveys showed lack of standardisation for vision assessment and treatment for stroke survivors. The National Stroke Strategy argues that vision and visual perceptual difficulties are components requiring multifaceted stroke-specific rehabilitation and support.[17] The Royal College of Physicians recommend that every patient with stroke should have a practical assessment of vision and examination of the visual field.[10]

Problems exist with referral accuracy from the MDT where there is suspected visual difficulty. It is reported that where referral by the MDT was based on the identification of ocular signs *only*, there was reduced sensitivity (42%) and specificity (52%).[3] Referral accuracy improved when visual symptoms were taken into account. Concerns were raised regarding potential failure to refer those patients unable to report their visual symptoms due to communication and cognitive deficits.[3] Inconsistencies between identification of ocular signs on assessment by the MDT and final ocular diagnosis have also been documented in an audit of stroke referrals for vision assessment.[18] Fifty-six per cent of visual diagnoses made prior to formal eye assessment were incorrect with amended

diagnoses being made following visual assessment by the orthoptic/ophthalmic team.[18] Our VISA screen at this early pilot stage appears to increase the accuracy of screening by increasing the ability to detect ocular signs separate from reporting of vision symptoms.

In each of the above studies, the MDT used a screening form on which they specified whether they noted any obvious visual signs such as nystagmus, strabismus or ptosis and whether the patient complained of visual symptoms such as double vision or reading difficulty. They did not, however, undertake any measurement of visual function. A further study evaluated Cardiff cards as a screening measure to identify low levels of vision.[19] A comparative study of qualitative methods of visual field assessment reported the difficulty in screening for visual field impairment in acute stages of stroke follow-up.[20] However, the authors recognised that confrontation is widely regarded as the most viable screening option for bedside visual field assessment.[19] Visual inattention is the most common visual perceptual disorder and there are various screening assessments in use for its detection but which do not extend to other facets of visual impairment.[21] In each of these studies, individual assessments of one aspect of visual function are considered. However, an overall visual screening assessment for stroke survivors is currently not available for use by MDTs in the absence of assessment by eye care professionals.[21]

## Limitations

The VISA screening tool was used by a combination of medial students and orthoptists while specialist vision assessment was provided by a team of orthoptists and ophthalmologists. Arguably, results wold be more meaningful if all VISA screens were completed by staff naïve to any vision assessment. Because this was a pilot validation study, we chose to include VISA screens from both medical students with no vision assessment experience and orthoptists who were experienced in vision assessment. Medical students represented completely naïve individuals in conducting vision screening assessments. However, orthoptists were chosen as screeners in this pilot stage of validation to serve as a quality check of the screening tool's ability to accurately assess various aspects of visual impairment. Our process evaluation for acceptability of the VISA screen involved feedback and interviewers with screeners only. We acknowledge this limitation and an important next step is to obtain views of stroke survivors on the acceptability of the VISA screen and its perceived value to them. A further limitation is that the VISA screen was not timed consistently for duration. Completion of the VISA screen was approximately 10 min in the small number that could be assessed but this cannot be taken as a representative screen duration. The screening duration is an important consideration when adding to busy acute stroke services and will be captured fully in the next stage of validation.

Our next stage of development is a full clinical validation of the VISA tool where all screening assessments are

completed by naïve screeners versus reference comprehensive vision assessment.

## CONCLUSIONS

This early validation of the VISA screening tool shows promise in improving detection accuracy for clinicians involved in stroke care who are not specialists in vision problems and lack formal eye training, with potential to lead to more prompt referral with fewer false positives and negatives. Clinicians reported acceptability of the VISA screening tool for use in screening for presence of vision problems in stroke survivors. Referral sensitivity of 90% and specificity of 80% were found for the VISA screening with strong inter-rater agreement for referral between VISA screening and specialist vision assessments.

The benefits are that the VISA screening tool may support increased speed of access to appropriate treatment of visual impairment and potential to preserve and make best use of remaining visual function for patients. Identification of visual impairment and implementation of early interventions and compensatory options have impact to overall rehabilitation, quality of life and activities of daily living with potential cost savings to the NHS by enhancing rehabilitation and supporting early discharge. Establishment of an effective vision screening tool is likely to be highly transferable to other vulnerable groups in other hospital inpatient areas, residential care settings or community MDT assessments.

**Acknowledgements** The authors express thanks to the staff and patients at Salford Royal Hospital and the Walton Centre.

**Contributors** All authors (FJR, LRH, KLH and CH) were involved in study set-up and data collection. LRH and FJR carried out the data analysis and wrote the initial manuscript draft. The manuscript was critically reviewed by KLH and CH. All authors read and approved the final manuscript.

**Funding** This study received no specific grant from any funding agency in the public, commercial or not-for- profit sectors. FJR and CH are in part funded by the National Institute for Health Research fellowship awards.

**Competing interests** None declared.

**Patient consent** Not required.

**Ethics approval** Office for Research Ethics Committees Northern Ireland (reference 16/NI/0125).

**Provenance and peer review** Not commissioned; externally peer reviewed.

**Data sharing statement** No additional data available.

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
