## [Reviewer comments · BMJ Open]

ARTICLE DETAILS

TITLE (PROVISIONAL)	Visual Impairment Screening Assessment (VISA) tool; pilot validation.
AUTHORS	Rowe, Fiona; Hepworth, Lauren; Hanna, Kerry; Howard, Claire

VERSION 1 – REVIEW

REVIEWER	Andoret van Wyk Department of Physiotherapy, Faculty of Health Sciences, University of Pretoria, Pretoria, South Africa
REVIEW RETURNED	14-Dec-2017

GENERAL COMMENTS	Thank you for the opportunity to review the article. Abstract Suggestion: Line 5 – To develop and evaluate Line 6: by the multidisciplinary stroke team Line 12: please add to the research setting (location) for example ... in the UK Line 15: Both the screening tool and comprehensive reference vision assessment noted case history and evaluated / assessed / observed visual acuity, eye alignment, eye movements, visual field and visual inattention. Page 3 of 16 Line 10: please add examples for various treatment options There are various treatment options such as that may be implemented / used that may have a beneficial effect on vision and to general rehabilitation. Line 15: I agree with the sentence (statement) but does not fit as it specifically refers to 'older persons' and not specifically to the post-stroke population (study population). The general post-stroke population is not limited to 'older persons'. Line 28 – 33: very long sentence, please revise. Also replace the word 'plus' Line 42: neuro-ophthalmologists Line 42: The panel considered results of recent stroke / vision research studies that utilised multiple measures of visual function such as / that included – please add examples of measures of visual function used in these studies. Line 46: please specify 'They' Line 46 – 49: please revise sentence Suggestion: visual impairments
---

	that may be present in patients post-stroke..... Page 4 of 16 Line 23: for inclusion in the study Line 26: please define severe cognitive impairment Was the potential patients' level of cognition formally screened or assessed with an outcome measure prior to the visual screening? Line 26: was aphasia patients excluded? Please provide more information of your exclusion criteria. Line 28 – 29: Please revise sentence ...and inclusive of as many stroke survivors admitted to the stroke units at the three hospital settings. [The stroke survivors were limited to the three hospital settings.] Line 52 – Line 57: Please elaborate on tests used during the comprehensive vision assessment, the tests used as reference during the full vision assessment Page 5 of 16 Line 12: <0.2 please add unit of measurement (specific test) Line 6 of 16 Line 13: replace 'by' Page 9 of 16 Line 23-24: please replace 'about' with approximately Page 10 of 16 Line 39: please replace 'Our' Line 42: increasing the ability to? detect Line 11 of 16 Line 47: please rephrase 'make best use' of residual visual function
--	--

REVIEWER	Kimberly Hreha University of Washington
REVIEW RETURNED	08-Jan-2018

GENERAL COMMENTS	I think that the authors need to clearly write the difference between the "screen" and "assessment". This is the first major problem. I like the push for comprehensive battery, that is routinely completed. I understand the authors concern, there is no standardized protocol for screening or referral. But there is assessments being completed, so why couldn't the authors push to have the assessment be completed routinely? Also, visual inattention, is not a visual disorder. Assessment for visual inattention should not be completed in a visual screening or assessment battery. Visual inattention, is not the word choice that this reviewer suggests should be used, but rather the authors could use the term "spatial neglect" and then mention that the condition manifests in the visual modality. Again, because the authors were vague in their explanation of the assessment and the screening battery, this reviewer is not sure what assessment is being used to assess for the visual inattention. It may not have been an appropriate assessment, which is what lead to the false positives or the lack of some of the assessments being completed. There are also questions that I have with the addition to the qualitative part of the study. This reviewer thinks that the authors did
---

	not explain this enough, and also this was not explained in either the abstract or the beginning of the paper. Also, the authors say that the primary outcome measure was the presence of or absence of visual impairment, but then the results are about validation? Including sensitivity, specificity and predictive values? This should be mentioned earlier. Then the authors later talk about test component agreement.
--	---

REVIEWER	Dr Katie Meadmore University of Southampton, UK
REVIEW RETURNED	08-Jan-2018

GENERAL COMMENTS	Comments to the Author Review of manuscript: bmjopen-2017-020562 entitled "Visual Impairment Screening Assessment (VISA) tool; pilot validation." This paper reports and evaluates a novel screening tool for visual impairment in stroke. It is a relevant and interesting topic, and one that I think would be of interest to BMJ open readers. However, there are a few issues that should be addressed. General: The term visual assessment is used to refer to the VISA and comprehensive vision assessment. As such, in places it got a bit confusing. Consider using VISA throughout for the developed tool. Abstract: Page 2, line 13 – please make it clear that the screeners are naïve/non-naïve to visual impairments. Page 2, line 29 – please clarify that the acceptability was from the screeners using the tool and not the stroke survivors being screened. Background: Page 3, line 6-10 – this sentence is difficult to follow. Consider re-wording. Method: Page 3, line 55 – it would be beneficial to have some additional detail here about how the final visual screening tool evolved from the first draft. How many sections did the tool start with? Were any sections added or removed? What type of feedback was provided? What was changed and revised due to this process? Were revisions mainly driven by healthcare professionals or stroke patients? Page 4 – the procedure needs clarification. It is not clear until the first paragraph of the results that only two assessments take place, and who undertook these assessments. In places the method makes it sound like the patients are screened on arrival to the unit and then completed the two study assessments. Please revise. Also it is not clear exactly what the participants experienced. Did the screening tools (the comprehensive assessment and VISA) list the different screening assessments in a particular order (and in which case what was this and what was the rationale behind this)? You also need to include a statement regarding ethical approval and informed consent. Page 5 – process evaluation. Please provide more details about this. When was this done? Who did the interviews? It is not clear if the
--

	process evaluation relates to the initial development of the tool or during the study. Page 5, line 53. What was the average time since stroke for the stroke participants who were screened? Page 6 – missing data. Please describe what was done with missing data. Were participants still included if they had missing data? Did missing data automatically mean a failure in that assessment section, or was the reason behind the failure taken into consideration? Table 1 – it would be helpful for those who are not familiar with the terminology to have descriptions of what true positive, false negative etc mean (e.g., visual impairment present and detected). Page 9 – line 23. Is genitive the right word? Page 9, line 35 – 40. This sentence is confusing. Page 9 – line 48-52. More detail here please. Which were the more interactive components? Also please clarify what is meant by “in such cases” – what cases? Page 10, line 26. Needs references Page 10, line 42. Doesn't read properly – typo? To detect not and detect? Page 11, line 42. Please make it clear that the VISA screening tool shows promise in improving detection accuracy for healthcare professionals involved in stroke care who are not specialists in visual impairments. Conclusions – do to state acceptability, which is described in the abstract conclusions. The authors should also consider an additional line or two relating to the actual findings rather than just the potential that a visual impairment tool like the VISA has.
--	--

VERSION 1 – AUTHOR RESPONSE

11th January 2018

Dear Adrian,

Manuscript ID bmjopen-2017-020562: Visual Impairment Screening Assessment (VISA) tool; pilot validation

Thank you for the opportunity to respond to the reviewer comments. We have provided our responses alongside individual comments below. Amendments are made in the revised manuscript as tracked changes.

Reviewer: Andoret van Wyk

Minor revisions recommended but not specified.

Reviewer: Kimberly Hreha

I think that the authors need to clearly write the difference between the "screen" and "assessment". These terms have been defined in the methods section and further distinction made throughout the paper between VISA screens and specialist vision assessments.

I understand the authors concern, there is no standardized protocol for screening or referral. But there is assessments being completed, so why couldn't the authors push to have the assessment be completed routinely?

It is not possible to push for routine completion of vision assessments as, currently, there are some stroke units that provide no assessment at all. In the UK and other countries, there are no/limited resources for any type of vision assessment – screen or otherwise. VISA was developed in the knowledge of these issues.

Visual inattention, is not a visual disorder. Assessment for visual inattention should not be completed in a visual screening or assessment battery. Visual inattention, is not the word choice that this reviewer suggests should be used, but rather the authors could use the term "spatial neglect" and then mention that the condition manifests in the visual modality.

We disagree that visual inattention should not be included in the VISA screen. When devising the VISA screen we followed a thorough consultation procedure to obtain multidisciplinary and patient views on the key assessments to include. It was a unanimous decision to include visual inattention. We have revised the wording to state spatial neglect with visual inattention as one modality.

Again, because the authors were vague in their explanation of the assessment and the screening battery, this reviewer is not sure what assessment is being used to assess for the visual inattention. It may not have been an appropriate assessment, which is what lead to the false positives or the lack of some of the assessments being completed.

Further detail of the screen assessments has been provided in the methods section.

There are also questions that I have with the addition to the qualitative part of the study. This reviewer thinks that the authors did not explain this enough, and also this was not explained in either the abstract or the beginning of the paper.

The process evaluation methods section has been expanded to provide further detail.

Also, the authors say that the primary outcome measure was the presence of or absence of visual impairment, but then the results are about validation? Including sensitivity, specificity and predictive values? This should be mentioned earlier. Then the authors later talk about test component agreement.

The statistical analysis section has been revised for clarity.

The objectives and the methodology needs to also be clearer.

Further revision has been made to clarify these sections.

Reviewer: Dr Katie Meadmore

General:

The term visual assessment is used to refer to the VISA and comprehensive vision assessment. As such, in places it got a bit confusing. Consider using VISA throughout for the developed tool.

We have amended the manuscript accordingly.

Abstract:

Page 2, line 13 – please make it clear that the screeners are naïve/non-naïve to visual impairments. A statement to this effect has been added.

Page 2, line 29 – please clarify that the acceptability was from the screeners using the tool and not the stroke survivors being screened.

This has been amended.

Background:

Page 3, line 6-10 – this sentence is difficult to follow. Consider re-wording.

We have reworded this sentence.

Method:

Page 3, line 55 – it would be beneficial to have some additional detail here about how the final visual screening tool evolved from the first draft. How many sections did the tool start with? Were any sections added or removed? What type of feedback was provided? What was changed and revised due to this process? Were revisions mainly driven by healthcare professionals or stroke patients?

Further information has been added in response to these points.

Page 4 – the procedure needs clarification. It is not clear until the first paragraph of the results that only two assessments take place, and who undertook these assessments. In places the method makes it sound like the patients are screened on arrival to the unit and then completed the two study assessments. Please revise. Also it is not clear exactly what the participants experienced. Did the screening tools (the comprehensive assessment and VISA) list the different screening assessments in a particular order (and in which case what was this and what was the rationale behind this)? You also need to include a statement regarding ethical approval and informed consent.

The recruitment and assessment information has been revised to clarify these details. The order of testing information has also been amended.

Page 5 – process evaluation. Please provide more details about this. When was this done? Who did the interviews? It is not clear if the process evaluation relates to the initial development of the tool or during the study.

Further details have been provided.

Page 5, line 53. What was the average time since stroke for the stroke participants who were screened?

We did not document precise data on time since stroke.

Page 6 – missing data. Please describe what was done with missing data. Were participants still included if they had missing data? Did missing data automatically mean a failure in that assessment section, or was the reason behind the failure taken into consideration?

This information has been added to the results section.

Table 1 – it would be helpful for those who are not familiar with the terminology to have descriptions of what true positive, false negative etc mean (e.g., visual impairment present and detected).

Amended accordingly.

Page 9 – line 23. Is genitive the right word?

Typo error – negative.

Page 9, line 35 – 40. This sentence is confusing.

The wording has been revised.

Page 9 – line 48-52. More detail here please. Which were the more interactive components? Also please clarify what is meant by “in such cases” – what cases?

This has been clarified.

Page 10, line 26. Needs references
References added.

Page 10, line 42. Doesn't read properly – typo? To detect not and detect?
Typo error which has been corrected.

Page 11, line 42. Please make it clear that the VISA screening tool shows promise in improving detection accuracy for healthcare professionals involved in stroke care who are not specialists in visual impairments.

This has been clarified.

Conclusions – do to state acceptability, which is described in the abstract conclusions. The authors should also consider an additional line or two relating to the actual findings rather than just the potential that a visual impairment tool like the VISA has.

These comments have been actioned.

Yours sincerely,

Prof Fiona Rowe
Professor of Orthoptics and Health Services Research
E: rowef@liverpool.ac.uk

VERSION 2 – REVIEW

REVIEWER	Kimberly Hreha University of Washington
REVIEW RETURNED	18-Jan-2018

GENERAL COMMENTS	I think that the authors did a great job with the revision. It is much clearer, more well defined and technical. I would just suggest to them to add this reference, because I think that their introduction section would benefit from having a sentence about theory: Roberts, P., Rizzo, J.R., Hreha, K., Wertheimer, J., Kaldenberg, J., Hironaka, D., Riggs, R., & Colenbrander, A. (2016). A conceptual model for vision rehabilitation. Journal of Rehabilitation Research and Development, 53 (6), 693-704.
--

REVIEWER	Katie Meadmore University of Southampton, UK
REVIEW RETURNED	22-Jan-2018

GENERAL COMMENTS	Typos: Background - penultimate sentence; assessments not assessment Conclusions - 2nd sentence: for its use not is use
--